# An Insight into the Bicarbonate Effect in Photosystem II through the Prism of the JIP Test

Alexandr V. Shitov

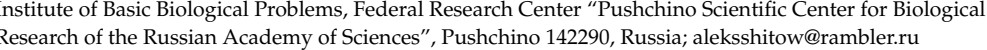

Institute of Basic Biological Problems, Federal Research Center "Pushchino Scientific Center for Biological Research of the Russian Academy of Sciences", Pushchino 142290, Russia; aleksshitow@rambler.ru

**Abstract:** Photosystem II (PSII) is the unique pigment–protein complex that is capable of evolving molecular oxygen using solar energy. The activity of PSII determines the overall productivity of all oxygenic photosynthetic organisms. It is well known that the absence of $HCO_3^-$ induces a drop in the activity of PSII. However, it is not yet clear what type of photochemical reaction, single turn-over or multiple turn-over, $HCO_3^-$ is involved in. Kinetic parameters of this (these) involvement(s) are almost unexplored now. This work addresses these issues. Using the JIP test, being the perspective noninvasive method for measuring PSII activity in plants, this paper describes how $HCO_3^-$ deficiency affects the electron transfer on the oxidizing as well as the reducing sides of PSII in thylakoids and in PSII preparations from the leaves of pea plants. $HCO_3^-$ was found to be simultaneously involved both in single turn-over and in multiple turn-over events ("dynamical processes"). Moreover, the involvement of $HCO_3^-$ in dynamical photochemical processes was revealed to be associated with both sides of PSII, being the rate limiting on the reducing side, which follows from obtained kinetic parameters. The involvement of $HCO_3^-$ in dynamical processes as the constant exchangeable ligand is discussed for both the electron donor and acceptor sides of PSII.

**Keywords:** photosystem II; bicarbonate effect; JIP test; O-J-I-P fluorescence transients

## 1. Introduction

Photosystem II (PSII) is the most interesting pigment–protein complex in thylakoids of all oxygenic photosynthesizers, since only PSII can oxidize water using solar energy to provide electrons for the electron transfer chain in thylakoids. Further, these electrons and the energy captured by both photosystems I and II produce the reducing power (in the form of NADPH and ATP) to synthesize carbohydrates [1,2]. Thus, PSII is the complex that determines the productivity of photosynthetic organisms. However, the detailed mechanism of the PSII function is not yet clear.

$HCO_3^-$ (BC) is one of the intriguing and important inorganic cofactors, which affects PSII photochemistry (the BC effect); it has been extensively studied over the past five decades (see reviews [3–5]). The deficiency of $HCO_3^-$ was found to decrease the rate of electron transfer on both the acceptor side [6–8] (where the first quinone acceptor ($Q_A$) and the second quinone acceptor ($Q_B$) are reduced) and on the donor side [4,9–12] (where water oxidation and oxygen evolution take place). However, the detailed mechanism of involvement of BC in photochemical reactions is not fully understood. On the one hand, it is generally accepted that $HCO_3^-$ acts in PSII as a permanent (non-replaceable) ligand between $Q_A$ and $Q_B$ near the non-heme Fe on the acceptor side [3,13–17]. On the other hand, recent data implies that $HCO_3^-$ may act as an exchangeable ligand. First, the binding constant of $HCO_3^-$ on the acceptor side was found to change during PSII function [18] which may indicate that BC could periodically dissociate from its binding site near the non-heme Fe. However, this proposal needs to be additionally tested by other methods. Second, there is the theory of "mobile" BC on the donor side of PSII [19–21]. $HCO_3^-$ was shown to stimulate $O_2$ evolution and electron transfer on the donor side

of PSII [10,12], while, in X-ray structures, it was not found as a tightly bound ligand on this side [14,15,17]. To explain this contradiction, $HCO_3^-$ was proposed to be a "mobile" ligand (i.e., a constantly interchangeable ligand), which accepts protons released during the process of photosynthetic water oxidation [19–21]. $CO_2$ was found to evolve along with $O_2$, resulting from the acceptance of $H^+$s by $HCO_3^-$, in accordance with the reaction $HCO_3^- + H^+ \leftrightarrow CO_2 + H_2O$. Nevertheless, these results contradict theoretical calculations. According to these calculations, if we assume that all $H^+$s released will be accepted by $HCO_3^-$, we will observe the release of $CO_2$, the amount of which should be four times more than the amount of $O_2$ evolved since the evolution of 1 mole of $O_2$ is accompanied by the release of 4 moles of $H^+$. Actually, in the work of Koroidov et al. [20] the release of $CO_2$ was ~60 times less than the evolution of $O_2$. Therefore, the hypothesis of "mobile" $HCO_3^-$ also needs to be tested by other approaches.

We ask: what approach could be used to test whether BC is an exchangeable ligand for chemical or photochemical reactions in PSII? In nature, BC exchanges very fast in reactions $HCO_3^- + H^+ \leftrightarrow CO_2 + H_2O$, catalyzed by the enzyme carbonic anhydrase (CA). The measurement of CA activity in PSII may help to answer the above question. Actually, the CA activity has been found by A. Stemler [22] and others [23–28] in PSII, indicating the involvement of BC in dynamical processes (associated with constant exchange of $HCO_3^-$) in PSII. Moreover, we have demonstrated that the CA activity of PSII provides a maximal electron transfer rate on the donor side of this complex [29,30]. Nevertheless, some researchers have doubted the important role of CA activity in the PSII function [31,32], since the measured rates of CA reactions in PSII were found to be less than the rate of $O_2$ evolution. Thus, studies of the CA activity in PSII cannot yet unambiguously determine whether the bicarbonate is a constantly exchangeable ligand in PSII.

The reason for low registered CA activity may be the low sensitivity of the methods used thus far. Therefore, to investigate whether BC is involved in dynamical processes in PSII, we need a new sensitive method that can register changes in the function of PSII with and without the presence of $HCO_3^-$. Measurement of photo-induced changes of chlorophyll a fluorescence yield related to the photoreduction of the PSII primary quinone electron acceptor, $Q_A$ (the variable chlorophyll fluorescence) may be this sensitive method. In addition, it is necessary that we measure changes in PSII function (for example, by measuring of electron transfer rates) on both sides of PSII simultaneously. The so-called JIP test [33], measuring fast variable chlorophyll *a* fluorescence, is the only approach that may satisfy these requirements.

The analysis of the fast fluorescence transients O-J-I-P by the JIP test [33] becomes very essential to control the condition of plants in a field. Under actinic light of high intensity (above 200 W m$^{-2}$ or 2500 μmol photons m$^{-2}$·s$^{-1}$), photosynthetic organisms exhibit chlorophyll (Chl) *a* fluorescence signals, which have specific changes that provide information about the structure and the function of the photosynthetic apparatus, especially, of PSII. The Chl *a* fluorescence rise exhibits usually the steps J (at 2–3 ms) and I (at 20–30 ms) between the initial O ($F_0$) and the maximum P ($F_P$) level, the entire transient being O-J-I-P. Here, $F_0$ is the minimum Chl a fluorescence measured at 50 μs, and $F_P$ is Chl a fluorescence intensity at the "P" level, the maximal fluorescence ($F_m$) [33–35]. In the JIP test, the J-step (whose fluorescence intensity is denoted as $F_j$) is taken as the border between single and multiple turn-overs in PSII [34]. The O-J part of the O-J-I-P fluorescence transient reflects single turn-over events, while the J-I-P part reflects multiple turn-over events [33], i.e., it is related to dynamical processes. Dynamical processes, described by J-I-P rise, may be related to events occurring on both sides of PSII. In particular, the J-I-P rise is generally correlated with the reduction of the plastoquinone (PQ) pool on the acceptor side [36,37], while the value of P, which is equal to $F_m$, may also include the effects of the donor side of PSII [10,29,30]. Thus, JIP test is able to provide information on dynamical processes simultaneously on both sides of PSII, although the nature of these dynamical processes remains unclear. Taking into account that BC is important for both sides of PSII, JIP test was found to have the potential of solving the question of how $HCO_3^-$ is involved

in photochemical processes in PSII. However, currently, no studies have been carried out directly on PSII preparations describing how $CO_2/HCO_3^-$ removal affects O-J-I-P fluorescence transients.

Thus, we ask: does BC take part in dynamical processes in PSII? To test it the following tasks were set. (i) To describe changes in O-J-I-P fluorescence transients arising from $CO_2/HCO_3^-$ removal from PSII preparations and thylakoid membranes. (ii) To find the location of the site(s) of these dynamical processes in PSII. (iii) To determine the kinetic parameters of $HCO_3^-$ mediated reactions.

This work describes the changes in O-J-I-P fluorescence transients, caused by the removal of $CO_2/HCO_3^-$ from either thylakoid membranes or PSII preparations isolated from Pea plants. $HCO_3^-$, actually, was found to take part in dynamical processes both on the acceptor and on the donor sides of PSII. Furthermore, two relatively independent dynamical processes, involving $HCO_3^-$, were revealed. The newly found changes in O-J-I-P transients under removal of $HCO_3^-$ may be used as diagnostic properties (as a "fingerprint" of BC effect) in the experiments in vivo (including in plants growing directly in a field) to reveal the inhibition of PSII by the deficiency of $HCO_3^-$.

## 2. Materials and Methods

### 2.1. Choice of Samples

Using the whole cells or chloroplasts, O-J-I-P transients reflect changes in activity not only of PSII, but also of Calvin–Benson–Bassham cycle [37]. Therefore, to investigate the BC effect directly on the PSII function, we must use PSII preparations that are free of the enzymes of the Calvin–Benson–Bassham cycle and, ideally, free of PSI also. BBY (Berthold–Babcock–Yocum) particles are good for this goal since they are free of the enzymes of Calvin–Benson–Bassham cycle and contain a relatively small amount of PSI. Nevertheless, the pool of PQ molecules in BBY is less than in thylakoid membranes, which may cause non-specific changes in O-J-I-P kinetics not directly related to the effect of BC. To exclude this non-specificity, we must study the effects of BC, using both BBY and thylakoid membranes. Therefore, in this study, we examined not only BBY but also thylakoid membranes.

Photochemically active thylakoid membrane preparations were prepared from leaves of pea plants (*Pisum sativum*) grown for 2–3 weeks by the method described in [38], with modifications, as in [39].

Photosystem II preparations (BBY-particles) were isolated from the thylakoid membrane preparations by the method described in [40].

### 2.2. Chlorophyll a Fluorescence Measurements

These measurements were carried out by monitoring photo-induced changes of Chl a fluorescence yield related to the photoreduction of the PSII primary quinone electron acceptor, $Q_A$ (the variable chlorophyll fluorescence) which mainly reflect the efficiency of electron transfer in PSII. The definitions of parameters of chlorophyll *a* fluorescence and the equations to calculate these parameters are presented in Table 1.

### 2.3. The Maximum Quantum Yield of the PSII Photochemistry

The $F_v/F_m$ ratio, which characterizes the maximum quantum yield of the PSII photochemistry in the dark-adapted samples, was measured as follows. $F_0$ and $F_m$ were measured in a 10-mm cuvette at 25 °C using an XE-PAM Pulse Modulated Fluorometer (Walz, Effeltrich, Germany). These measurements were carried out by suspending the samples in a medium containing 50 mM MESNaOH (pH 6.5 or pH 5.5) and 35 mM NaCl. The chlorophyll concentration was 10 $\mu g \cdot mL^{-1}$. The following illumination conditions were used: to record the $F_0$ level of fluorescence, the dark-adapted sample was illuminated by a weak probe pulse of measuring light ($\lambda$ = 490 nm; 4 $\mu$mol photons $m^{-2} s^{-1}$; Xenon-Measuring Flash Lamp, 64 Hz (Hamamatsu Photonics K.K., Hamamatsu, Japan), optical filter BG39 (Schott, Mainz, Germany). The level of $F_0$ was recorded at $\lambda > 660$ nm. For measurement of the photoinduced changes of the variable PSII chlorophyll fluorescence

yield ($F_v$), the dark-adapted samples were illuminated with the light of saturating intensity (1000 μmol photons m$^{-2}$ s$^{-1}$).

### 2.4. The JIP Test

The fast Chl a fluorescence rise kinetics (O-J-I-P) were measured using a Multi-Color PAM fluorometer (Walz, Germany). The intensity of the weak measuring light (λ = 490 nm) was 0.0618 μmol photons m$^{-2}$ s$^{-1}$, and the intensity of red actinic light (λ~650 nm) was 2.250 μmol photons m$^{-2}$ s$^{-1}$. $F_0$ was measured at 0.05 ms. All measurements were performed in 10 mm × 10 mm quartz cuvette (Hellma Analytics) at 25 °C with samples suspended in the same medium as was used in the measurements of the $F_v/F_m$ ratio and at Chl concentration of 10 μg·mL$^{-1}$. All samples were dark-adapted for at least 1 h. Before the measurement, samples were adapted to the corresponding buffer for 2 min in darkness.

### 2.5. Calculations Related to the JIP Test

The JIP test parameters, the abbreviations, all the formulas, and definitions of all symbols (cf. [33]) used in the current study are presented also in Table 1. The O-J-I-P fluorescence transients were normalized by using either: (1) a simple normalization at $F_0$, (where $F_0$ = 0); or (2) a double normalization, i.e., at both $F_0$ = 0 and $F_P$ ($F_m$) = 1. The rates of fluorescence transients in the range of "a", "b", and "c" were calculated as a tangent of slopes, shown by dashed lines in Figures 1A and 2A.

**Table 1.** The abbreviations, formulas, and definitions of the fluorescence (JIP test) parameters used in the current study.

| Abbreviations | Formulas | Definitions of The JIP Test Parameters |
|---|---|---|
| Area | The area between the fluorescence curve and the line F = $F_m$ | The total area over the O-J-I-P curve |
| $F_0$ | F (50 μs) | The initial level of the chlorophyll a fluorescence (fluorescence at 50 μs) |
| $F_m$ ($F_P$) | | The value of the maximum level of the fluorescence ($F_P$ for the JIP test) |
| $F_v$ | $F_v = F_m - F_0$ (1) | The value of photo-induced changes of the Chl a fluorescence yield related to the photoreduction of the PSII primary quinone electron acceptor, $Q_A$ (the variable chlorophyll fluorescence) |
| $F_v/F_m$ | | The maximum quantum yield of the PSII photochemistry in the dark-adapted samples |
| F(t) | | The value of photo-induced changes of the Chl a fluorescence yield at time t |
| $F_j$ | | The value of photo-induced changes of the Chl a fluorescence yield at the step J (at 2–5 ms) |
| $\tau F_j$ | | The time of reaching of $F_j$, ms |
| V(t) | $V(t) = (F(t) - F_0)/(F_m - F_0)$ (2) | The relative variable fluorescence yield at time t (with values between zero and 1) in a double normalized O-J-I-P kinetics, i.e., at $F_0$ = 0 and $F_P$ ($F_m$) = 1 |
| $V_j$ | $V_j = (F_j - F_0)/(F_m - F_0)$ (3) | The relative variable fluorescence yield at 2–5 ms in a double normalized O-J-I-P kinetics |
| 1-$F_j$ | $1-F_j = 1 - (F_j - F_0)/(F_m - F_0)$ (4) | The probability of electrons to move into the electron transfer chain further than $Q_A$ for a double normalized O-J-I-P kinetics |
| **N** | $N = [Area/(F_m - F_0)] \times M_0 \times (1/V_j)$ (5) | The turn-over number of $Q_A$ reduction events between time 0 to $F_m$ for a double normalized O-J-I-P kinetics |
| $M_0$ | $M_0 = [F_{(0.3\,ms)} - F_0]/(F_m - F_0)$ (6) | The initial slope of the O-J-I-P curve (slope of the O to J rise) for a double normalized O-J-I-P kinetics |
| $F_{(0.3\,ms)}$ | | The value of photo-induced changes of the Chl a fluorescence yield (the value of the fluorescence) at 0.3 ms |
| $R_a$ | $R_a = [F_{(2.5\,ms)} - F_{(1\,ms)}]/\Delta t$ (7) | The rate of the fluorescence rises at the time range of 1–2.5 ms (which is marked as "the range a" in Figures 1A and 2A), were $\Delta t$ = 1.5 ms |
| $F_{(2.5\,ms)}$ | | The value of the fluorescence at 2.5 ms |
| $F_{(1\,ms)}$ | | The value of the fluorescence at 1.0 ms |

| Abbreviations | Formulas | Definitions of The JIP Test Parameters |
|:---:|:---:|:---:|
| $R_b$ | $R_b = [F_{(30\ ms)} - F_{(6\ ms)}]/\Delta t$ (8) | The rate of the fluorescence rises at the time range of 6–30 ms (which is marked as "the range b" in Figures 1A and 2A), were $\Delta t$ = 24 ms |
| $F_{(30\ ms)}$ | | The value of the fluorescence at 30 ms |
| $F_{(6\ ms)}$ | | The value of the fluorescence at 6 ms |
| $R_c$ | $R_c = [F_{(800\ ms)} - F_{(400\ ms)}]/\Delta t$ (9) | The rate of the fluorescence rises at the time range of 400–800 ms (which is marked as "the range c" in Figures 1A and 2A), were $\Delta t$ = 400 ms |
| $F_{(800\ ms)}$ | | The value of the fluorescence at 800 ms |
| $F_{(400\ ms)}$ | | The value of the fluorescence at 400 ms |

### 2.6. Oxygen-Evolving Activity

The oxygen-evolving activity of BBY particles was measured amperometrically on the Clarke-type electrode at 1000 µmol photons m$^{-2}$ s$^{-1}$ of red light ($\lambda$ > 590 nm), in a 1 mL cell (Hansatech Instruments Ltd., Norfolk, UK), with samples suspended in a medium containing 50 mM MES-NaOH (pH 6.5), 35 mM NaCl and 300 mM sucrose. The chlorophyll concentration in the cell was 10 µg·mL$^{-1}$. All measurements were carried out at 25 °C in the presence of 0.3 mM 2,6-dichloro-pbenzoquinone and 1 mM potassium ferricyanide as electron acceptors.

### 2.7. Removal of Bicarbonate

As mentioned in the "Introduction", the main goal of this work is to describe how $CO_2/HCO_3^-$ deficiency affects the O-J-I-P fluorescence transients in PSII, allows us to know its site of action. There are two main approaches for the removal of $CO_2/HCO_3^-$ from the samples: (i) to replace $HCO_3^-$ from the sites of its action (for example, by formate); (ii) to dilute the concentrated sample with a medium depleted of endogenous $CO_2/HCO_3^-$. The first approach reveals BC effect not simultaneously for both sides of PSII, since the donor side needs 250–1000 fold less concentration of formate than the acceptor side [10]. The second approach is much more appropriate for the goal of this work since it removes BC from both sides of PSII simultaneously [4,9–12].

Partial removal of bicarbonate from samples was obtained by 800-fold dilution of concentrated (8.5 mg mL$^{-1}$) samples into a medium depleted of endogenous $CO_2/HCO_3^-$ by 60-min flushing with air (which was freed from $CO_2$ by passage through a solution of NaOH and ascarite) [11]. The sample was subsequently incubated in the medium for 2 min at 4 °C.

The content of $HCO_3^-$ additionally decreases by dilution of samples in the buffer with pH 5.5 instead of pH 6.5 (from 57.5% of $HCO_3^-$ at pH 6.5 to 12% of $HCO_3^-$ at pH 5.5), which provides a very low concentration of BC in the samples. To observe the effect of BC removal step by step, 3 types of solutions were used: (1) 50 mM MES-NaOH buffer (pH 6.5) that was not depleted of $CO_2/HCO_3^-$; this solution contained 288 µM $HCO_3^-$. (2) 50 mM MES-NaOH buffer (pH 5.5), which was not depleted of $CO_2/HCO_3^-$; this solution contained 38 µM $HCO_3^-$. (3) 50 mM MES-NaOH buffer (pH 5.5) which was depleted of $CO_2/HCO_3^-$; this solution contained 5 µM $HCO_3^-$. The specificity of BC effect was tested by the addition of 3 mM of $NaHCO_3$ to solution 3 (the obtained solution is marked as solution 4). This solution contained 365 µM $HCO_3^-$. As it has been found previously [30], 3 mM of extrinsically added BC was enough to observe a clear BC effect, but this concentration of BC did not induce pH shift if 50 mM concentration of the buffer was used. Thus, the O-J-I-P transients were measured in all samples using the system of 4 solutions mentioned above. All samples were incubated in corresponding solution in the dark for 2 min before the measurement.

The total content of carbon in non-organic forms ($CO_2$, $HCO_3^-$, $CO_3^{2-}$) was determined in solutions by measuring the amount of $CO_2$ that evolved from 1 mL of buffer after

the addition of 2 mL 5 M HCl; this was performed in a closed vial using a fixed known volume of the sample. The concentration of $CO_2$ was measured by a gas chromatography system KristalLux 4000 (Meta Chrom, Yoshkar-Ola, Russia).

### 2.8. Statistical Analysis

Measurements of Chl *a* fluorescence were statistically analyzed using ANOVA model. Only measurements having significant values ($p < 0.05$) are shown in the figures. Statistical analysis of results was performed by using standard algorithms of OriginPro 6.5 software package (OriginLab, Northampton, MA, USA). We used at least three replications for each measurement. The mean values and standard deviations were used for the plot curves (and the same were entered in the tables).

### 2.9. Chlorophyll Concentration

The total Chl content, in the preparations, was determined by the method of Porra et al. [41]. Buffered aqueous acetone (80%) was used as the solvent; this solvent contained 2.5 mM sodium phosphate buffer at pH 7.8.

### 2.10. The Concentration of Photochemical Reaction Centers of PSII

The concentration of photochemical reaction centers of PSII (RCII) was determined according to published methods [42,43]. In BBY preps, Chl concentration was 10 $\mu g \cdot mL^{-1}$, which was used for the JIP test measurements; it corresponded to having $6.08 \times 10^{-5}$ mM RCII ($[E_t]$).

### 2.11. Calculation of Kinetic Parameters ($V_{max}$, $K_m$, and $k_{cat}$)

Lineweaver–Burk plots [44] ($1/R$ versus $1/C$) were used to determine $V_{max}$ and $K_m$, where R is the rate of the reaction, and C is the substrate ($HCO_3^-$) concentration. The obtained plots were then used to derive their equations in the Origin program. Then the derived equations were used to calculate $K_m$ and $V_{max}$ values. $k_{cat}$ was calculated according to the following equation:

$$k_{cat} = V_{max}/[E_t] \tag{10}$$

where $V_{max}$ is the maximal rate, and $[E_t]$ is the concentration of RCII.

## 3. Results

### 3.1. The JIP Test in Thylakoids

The rate of $O_2$ evolution, in thylakoids, was equal to $289 \pm 14$ $\mu$mol $O_2 \cdot$(mg of Chl)$^{-1} \cdot h^{-1}$ and $F_v/F_m$ was equal to $0.72 \pm 0.02$ showing high photosynthetic efficiency of the thylakoids in our experiments. Thus, results from this sample could be used for the analysis of O-J-I-P transients. $F_P$ (which equaled $F_m$) was maximal when the sample was diluted in solution 1 (pH 6.5, which is optimal for PSII photochemical reactions). This fact also indicates that the intensity of illumination used in all experiments was saturating. Therefore, the illumination conditions used were sufficient for the JIP test in all our experiments (for results, see Figure 1).

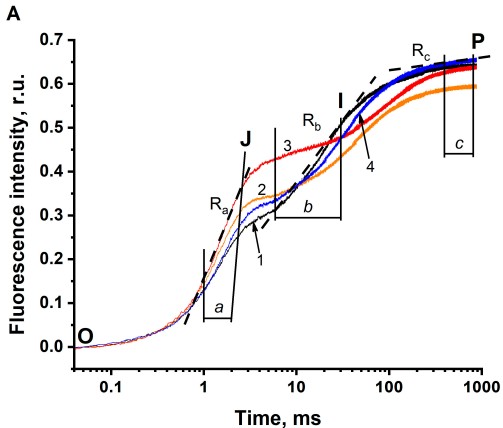

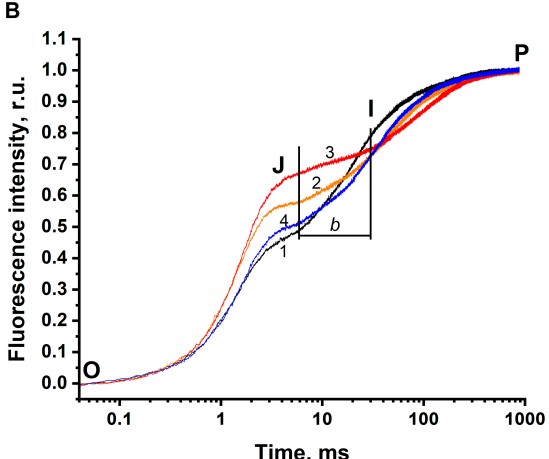

**Figure 1.** Effect of the removal of $HCO_3^-$ on the fast kinetics of the photo-induced changes of chlorophyll *a* fluorescence (the JIP test) in thylakoids. (**A**) All O-J-I-P kinetics were normalized to $F_0 = 0$. (**B**) All O-J-I-P kinetics were normalized to $F_0 = 0$ and to $F_P$ ($F_m$) = 1. Curve 1 (black) shows O-J-I-P kinetic in the buffer with pH 6.5, non-depleted of $CO_2/HCO_3^-$ (the concentration of $HCO_3^-$ in the solution was 288 μM). Curve 2 (orange) indicates O-J-I-P kinetic in the buffer with pH 5.5, non-depleted of $CO_2/HCO_3^-$ (the concentration of $HCO_3^-$ in the solution was 38 μM). Curve 3 (red) demonstrates O-J-I-P kinetic in the buffer with pH 5.5, depleted of $CO_2/HCO_3^-$ (the concentration of $HCO_3^-$ in the solution was 5 μM). Curve 4 (blue) shows O-J-I-P kinetic in the buffer with pH 5.5, depleted of $CO_2/HCO_3^-$ in the presence of 3 mM $HCO_3^-$ added exogenously (the concentration of $HCO_3^-$ in the solution was 365 μM). The range "a" shows the fluorescence rise in the range of 1–2.5 ms. The range "b" indicates the fluorescence rise in the range of 6–30 ms. The range "c" shows the fluorescence rise in the range of 400–800 ms. The rates in these ranges were calculated at linear sections of each kinetic (examples of these lines marked by broken lines). Thylakoid membranes were dissolved in 50 mM MES-NaOH buffer. The final concentration, in terms of Chl, of thylakoids was 10 μg·mL$^{-1}$. The measurements were repeated 3–6 times with similar results.

The removal of $CO_2/HCO_3^-$ from the sample induced clear changes in the O-J-I-P transients (see Figure 1A). The double normalized O-J-I-P kinetics are shown in Figure 1B. All changes are summarized as follows: (i) $\tau F_j$ shifts to longer times; (ii) $F_j$ increases and $1-F_j$ decreases; (iii) $F_P$ decreases (by comparing the sample in solution 1, having maximal $HCO_3^-$ concentration, with the same sample in solution 2, having ~8 times less concentration of $HCO_3^-$); (iv) $R_a$ increases and $1-R_a$ decreases ($R_a$ is the rate of fluorescence rise at the range of 1–2.5 ms (which is marked as "the range a"); (v) $R_b$ decreases ($R_b$ is the rate of fluorescence rise at the range of 6–30 ms (which is marked as "the range b"); (vi) $R_c$

increases and $1-R_c$ decreases ($R_c$ is the rate of fluorescence rise at the range of 400–800 ms (which is marked as "the range c").

Changes in the parameters of O-J-I-P transients are presented in Table 2. Remarkably, the increase in $F_j$ (the decrease in $1-F_j$) is seen more noticeably in the double normalized Figure 1B, as compared to that in Figure 1A. It is well known that O-J-I-P transients reflect the electron transfer rates at different parts of the electron transfer chain in PSII. Therefore, the changes in O-J-I-P kinetics, induced by the removal of $CO_2/HCO_3^-$, may reflect the involvement of BC in electron transfer reactions in PSII.

As shown in Figure 1A, the parameters $1-F_j$, $F_P$, $1-R_a$, $R_b$, and $1-R_c$ decreased (Table 2), when the concentration of BC decreased from 288 µM at pH 6.5 (Figure 1A, line 1) to 38 µM at pH 5.5 in the buffer that was not depleted of $CO_2/HCO_3^-$ (Figure 1A, line 2). If the concentration of $HCO_3^-$ was further decreased to 5 µM by flushing of solution 2 (for further information see "Materials and Methods"), all the above-mentioned parameters were found to further decrease (Figure 1A, line 3; Table 2). So, the decrease in these parameters occurs due to the decrease in the concentration of $HCO_3^-$ in the sample. The addition of BC (Figure 1A, line 4) reversed all the changes in O-J-I-P transients (Table 2), that was due to the removal of BC. This result indicates that the removal of $CO_2/HCO_3^-$ is actually the reason for all the changes in PSII photochemistry reflected in the O-J-I-P kinetics.

**Table 2.** Characteristics of O-J-I-P fluorescence transients in thylakoid membranes at different concentrations of $HCO_3^-$ in the buffer.

| Experimental Conditions | Characteristics of the O-J-I-P Fluorescence Transients | | | | | | | | |
|---|---|---|---|---|---|---|---|---|---|
| | $\tau F_j$ | $F_j$ | | $F_P$ | $1-R_a$ | | $R_b$ | | $1-R_c$ |
| | | | $1-F_j$ | | | | | | |
| | Norm. to $F_0$ | Norm. to $F_0$ | Norm. to $F_0$ and $F_P$ | Norm. to $F_0$ and $F_P$ | Norm. to $F_0$ | Norm. to $F_0$ | Norm. to $F_0$ | Norm. to $F_0$ and $F_P$ | Norm. to $F_0$ |
| Solution 1 pH 6.5 not depleted of $CO_2/HCO_3^-$ (288 µM $HCO_3^-$) | 2.9 ms (100%) | 0.28 (100%) | 0.43 (100%) | 0.57 (100%) | 0.65 (100%) | 0.654 ± 0.011 (100%) | 0.299 ± 0.009 (100%) | 0.478 ± 0.023 (100%) | 0.988 ± 0.003 (100%) |
| Solution 2 pH 5.5 not depleted of $CO_2/HCO_3^-$ (38 µM $HCO_3^-$) | 3.8 ms (131%) | 0.34 (121%) | 0.57 (133%) | 0.43 (75%) | 0.59 (91%) | 0.570 ± 0.008 (87%) | 0.127 ± 0.002 (42%) | 0.210 ± 0.010 (44%) | 0.975 ± 0.001 (99%) |
| Solution 3 pH 5.5 depleted of $CO_2/HCO_3^-$ (5 µM $HCO_3^-$) | 4.8 ms (166%) | 0.42 (150%) | 0.66 (153%) | 0.34 (60%) | 0.64 (98%) | 0.464 ± 0.005 (71%) | 0.064 ± 0.003 (21%) | 0.100 ± 0.003 21%) | 0.964 ± 0.002 (98%) |
| Solution 4 pH 5.5 depleted of $CO_2/HCO_3^-$ + 3 mM of $HCO_3^-$ (365 µM $HCO_3^-$) | 4.2 ms (145%) | 0.32 (114%) | 0.50 (116%) | 0.50 (88%) | 0.65 (100%) | 0.593 ± 0.004 (91%) | 0.190 ± 0.01 (64%) | 0.298 ± 0.009 (62%) | 0.972 ± 0.001 (98.4%) |

As it is seen in Figure 1A, the removal of $CO_2/HCO_3^-$ affected not only the parameters related to quasi-stationary states ($\tau F_j$, $F_j$ or $1-F_j$, and $F_P$) but also the rate of fluorescence rises $R_a$, $R_b$, and $R_c$ (Figure 1A). The $R_a$ along with $F_j$ or $1-F_j$, belonging to the O-J part, may reflect on the single turn-over events in PSII, while $R_b$ and $R_c$ along with the value $F_P$, belonging to the J-I-P part, may be related to the multiple turn-over events in PSII. Thus, these observations indicate that BC may be involved not only in quasi-stationary events (single turn-over events), but also in dynamical (multiple turn-over) events during the functioning of PSII.

### 3.2. The JIP Test in BBYs (Photosystem II)

The above-described JIP test on thylakoids may reflect changes in electron transfer not only in PSII but also on the acceptor side of photosystem I (PSI). So, the I-P rise of O-J-I-P transients may have been affected by the transient dark inactivation of ferredoxin-NADP$^+$-reductase and of the Calvin–Benson–Bassham cycle [37,45,46]. Therefore, to investigate the BC effect directly on PSII, the effects of BC on BBY preparation were examined.

The observed rate of $O_2$ evolution equal to $729 \pm 19$ μmol $O_2 \cdot$(mg of Chl)$^{-1} \cdot$h$^{-1}$ and $F_v/F_m$ equal to $0.74 \pm 0.01$ shows high photosynthetic efficiency of the BBYs, used in this research. It indicates that this sample may indeed be used for further analysis of O-J-I-P transients. $F_P$ (which equaled $F_m$) was found to be maximal if the sample was diluted in solution 1 (pH 6.5, which is optimal for PSII photochemistry). This fact indicates that the intensity of illumination used in all our experiments was saturating. Therefore, the illumination conditions used were sufficient for the JIP test in all experiments.

The effect of BC on O-J-I-P transients in BBYs was investigated in the same way used for experiments on thylakoid membranes. O-J-I-P transients obtained with BBYs are shown in Figure 2A. The O-J-I-P kinetics normalized to $F_0 = 0$ and to $F_P$ ($F_m$) = 1 are presented in Figure 2B.

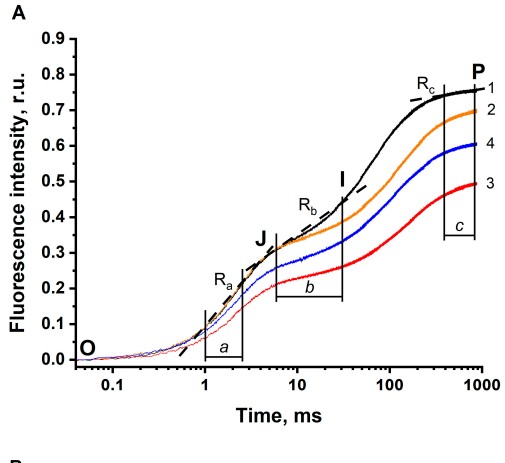

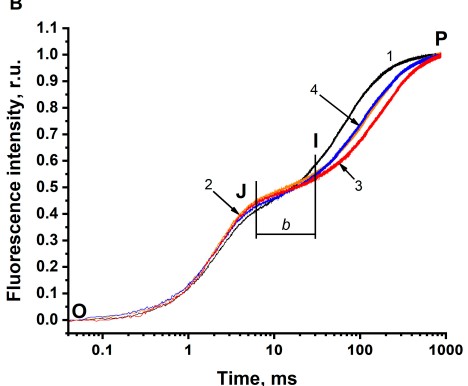

**Figure 2.** Effect of the removal of $HCO_3^-$ on the fast (OJIP) kinetics of the photo-induced changes of chlorophyll fluorescence (JIP test) in BBYs. (**A**) All O-J-I-P kinetics were normalized to $F_0 = 0$. (**B**) All O-J-I-P kinetics were normalized to $F_0 = 0$ and to $F_P$ ($F_m$) = 1. Curve 1 (black) shows O-J-I-P kinetics in the buffer with pH 6.5, non-depleted of $CO_2/HCO_3^-$ (the concentration of $HCO_3^-$ in the solution was 288 μM). Curve 2 (orange) indicates O-J-I-P kinetics in the buffer with pH 5.5, non-depleted of $CO_2/HCO_3^-$ (the concentration of $HCO_3^-$ in the solution was 38 μM). Curve 3 (red) is for the O-J-I-P kinetics in the buffer with pH 5.5, depleted of $CO_2/HCO_3^-$ (the concentration of $HCO_3^-$ in the solution was 5 μM). Curve 4 (blue) shows the O-J-I-P kinetics in the buffer with pH 5.5, depleted of $CO_2/HCO_3^-$ in the presence of 3 mM $HCO_3^-$ added exogenously (the concentration of $HCO_3^-$ in the solution was 365 μM). The range "a" shows the fluorescence rise during 1–2.5 ms. The range "b" indicates the fluorescence rise during 6–30 ms. The range "c" shows the fluorescence rise during 400–800 ms. The rates in these ranges were calculated from the linear sections of each kinetics (examples are marked by broken lines). BBYs were dissolved in 50 mM MES-NaOH buffer. The final concentration of BBYs, in terms of Chl, was 10 μg·mL$^{-1}$. The measurements were repeated 3–6 times with similar results.

All fluorescence parameters previously searched in thylakoids were also determined for BBY. The obtained results are presented in Table 3.

**Table 3.** Characteristics of O-J-I-P fluorescence transients in BBYs at different concentrations of $HCO_3^-$ in the buffer.

| Experimental Conditions | $\tau F_j$ | $F_j$ | | $1-F_j$ | $F_P$ | $R_a$ | $R_b$ | | $1-R_c$ |
|---|---|---|---|---|---|---|---|---|---|
| | Norm. To $F_0$ | Norm. To $F_0$ | Norm. to $F_0$ and $F_P$ | Norm. to $F_0$ and $F_P$ | Norm. To $F_0$ | Norm. To $F_0$ | Norm. To $F_0$ | Norm. to $F_0$ and $F_P$ | Norm. To $F_0$ |
| Solution 1 pH 6.5 not depleted of $CO_2/HCO_3^-$ (288 µM $HCO_3^-$) | 3.7 ms (100%) | 0.27 (100%) | 0.35 (100%) | 0.65 (100%) | 0.76 (100%) | 0.316 ± 0.005 (100%) | 0.187 ± 0.012 (100%) | 0.249 ± 0.011 (100%) | 0.961 ± 0.004 (100%) |
| Solution 2 pH 5.5 not depleted of $CO_2/HCO_3^-$ (38 µM $HCO_3^-$) | 4.1 ms (111%) | 0.27 (100%) | 0.43 (123%) | 0.57 (88%) | 0.70 (92%) | 0.315 ± 0.007 (100%) | 0.101 ± 0.007 (54%) | 0.145 ± 0.009 (58%) | 0.914 ± 0.002 (95%) |
| Solution 3 pH 5.5 depleted of $CO_2/HCO_3^-$ (5 µM $HCO_3^-$) | 4.8 ms (130%) | 0.20 (74%) | 0.41 (117%) | 0.59 (91%) | 0.50 (66%) | 0.217 ± 0.006 (69%) | 0.065 ± 0.002 (35%) | 0.131 ± 0.004 (53%) | 0.903 ± 0.001 (94%) |
| Solution 4 pH 5.5 depleted of $CO_2/HCO_3^-$ + 3 mM of $HCO_3^-$ (365 µM $HCO_3^-$) | 3.9 ms (105%) | 0.23 (85%) | 0.38 (109%) | 0.62 (95%) | 0.61 (80%) | 0.259 ± 0.004 (82%) | 0.102 ± 0.003 (55%) | 0.170 ± 0.007 (68%) | 0.929 ± 0.002 (97%) |

The removal of $CO_2/HCO_3^-$ from the sample induced clear changes in the parameters of O-J-I-P transients. These changes are associated with the BC effect since the addition of 3 mM BC (Figure 2A, curve 4) reverses all the changes in O-J-I-P transients (Table 3, line 4), that had resulted from the removal of BC. Therefore, the difference (by percentage) between a parameter in solution 1 (where BC concentration was maximal (288 µM)) and the corresponding parameter in solution 3 (where BC concentration was minimal (5 µM)), hereinafter, will be denoted as "the magnitude of the BC effect". All the changes in parameters of O-J-I-P transients are summarized below:

1. $\tau F_j$ shifts to longer times. In BBYs, this shift was found to be less pronounced (the magnitude of the BC effect was 30%) than in the thylakoids (this magnitude equaled 66%). Remarkably, in BBY, $\tau F_j$ has longer times, as compared to that in thylakoids, even under optimal conditions (pH 6.5 in the buffer non-depleted of $CO_2/HCO_3^-$).

2. $F_j$ increases (if $F_j$ was double normalized) and $1-F_j$ decreases. The magnitude of the BC effect was found to be less in BBY (23 and 12%, respectively), compared to the thylakoids (53 and 40%, respectively (cf. Tables 2 and 3, columns 3 and 4, respectively)).

3. $F_P$ clearly decreases being more pronounced in BBY (the magnitude of the BC effect was equal to 34%), compared to that in the thylakoids (where the magnitude was only 9%, at least when comparing $F_P$ in solution 2) (see Table 3, column 5). This means that the BC effect on $F_P$ is seen more clearly in BBYs compared to that in the thylakoids.

4. The $R_a$ decreases. The magnitude of this decrease in BBY was similar to the decrease in $1-R_a$ in thylakoids.

5. The $R_b$ decreases, which was similar to that of thylakoids.

6. The $R_c$ increases and the $1-R_c$ decreases. The decrease in $1-R_c$ was found to be more pronounced in BBY (the magnitude of the BC effect was equal to 6%), compared to that in thylakoids (where the magnitude was only 2%).

In general, the removal of $CO_2/HCO_3^-$ results in similar changes in O-J-I-P transients in BBYs and in thylakoids. The only exception was $R_a$; it was found to decrease in BBY, but to increase in thylakoids. The reason for this phenomenon is not yet clear, and additional experiments are required to establish it. Therefore, we have analyzed kinetic parameters for only the $R_b$ and the $1-R_c$, since they changed in a similar manner both in BBYs and in thylakoids.

### 3.3. Kinetic Parameters of O-J-I-P Transients for the Rates $R_b$ and 1-$R_c$ in Thylakoid Membranes and in PSIIs (BBYs)

Observing the effect of BC on the $R_b$ and on the 1-$R_c$, the question arises: are these rates the result of one process, or do they reflect two independent processes? To answer this question, the kinetic parameters of these processes were determined using the Lineweaver and Burk plots (1/R vs 1/C), where R is the $R_b$ and 1-$R_c$, respectively, and C is the concentration of $HCO_3^-$ in solutions. The corresponding plots for thylakoids are given in Figures S1 and S2, and the plots for BBY are in Figures S3 and S4. The obtained kinetic parameters are presented in Table 4. As it is seen from Table 4, kinetic parameters for the $R_b$ differ from that of the 1-$R_c$–both in thylakoids and in BBYs. This indicates that, on the one hand, BC functions differently in processes described by the $R_b$ and the 1-$R_c$, but on the other hand, BC is involved in these processes almost simultaneously.

**Table 4.** Kinetic parameters of dynamical processes involving $HCO_3^-$ described by the rates $R_b$ and 1-$R_c$ in O-J-I-P transients for thylakoid membranes and for BBY.

| Kinetic. Parameters | Thylakoids | | BBY | |
|---|---|---|---|---|
| | **For $R_b$** | **For 1-$R_c$** | **For $R_b$** | **For 1-$R_c$** |
| $K_m$, mM | $11.4 \times 10^{-3}$ | $7.4 \times 10^{-5}$ | $3.9 \times 10^{-3}$ | $2.0 \times 10^{-4}$ |
| $V_{max}$, rel.un. | 0.207 | 0.980 | 0.124 | 0.939 |
| $k_{cat}$, $s^{-1}$ | - | - | $2.0 \times 10^6$ | $1.6 \times 10^7$ |

To find $k_{cat}$ one needs to know the concentration of RCII (for more information see Equation (10) in "Materials and Methods"). This was not possible in thylakoids because there is no simple and reproducible method to determine the concentration of RCII directly. The simple method has been described only for BBY preparations [42]. Therefore, $k_{cat}$ for the $R_b$ and for the 1-$R_c$ could be determined only in the BBYs.

$K_m$ for the $R_b$ was found to be three times less in BBY than that in the thylakoids, which indicates a higher affinity to BC for reactions described by the $R_b$ in BBY. $V_{max}$ for the $R_b$ was found to be ~2 times less in BBY than that in thylakoids, which shows lower efficiency of reactions described by the $R_b$ in BBY. $K_m$ for the 1-$R_c$ was found to be ~3 times more in BBY than that in thylakoids, which indicates less affinity to BC for reactions described by the 1-$R_c$ in BBY. $V_{max}$ for the 1-$R_c$ was found to be very similar in BBY and in thylakoids, which shows similar efficiency of the reactions described by the 1-$R_c$ in both preparations. The $k_{cat}$ for the 1-$R_c$ was found to be almost an order of magnitude higher than $k_{cat}$ for the $R_b$, indicating that the process(es) described by the 1-$R_c$ completely differ(s) from the process(es) described by the $R_b$. Thus, the $R_b$ may describe a type of photochemical process(es) involving $HCO_3^-$ that differs from the photochemical process(es) described by the 1-$R_c$ also involving $HCO_3^-$. In addition, these processes occur almost simultaneously in PSII.

Taking into account that $V_{max}$ and $k_{cat}$ for the $R_b$ was less than these parameters for the 1-$R_c$, the process(es) described by the $R_b$ seems to be rate-limiting for the electron transfer through PSII. However, this proposal needs to be verified by other approaches.

We now ask: can the processes described by the $R_b$ and the 1-$R_c$ correlate with real photochemical processes in PSII? There is one real photochemical process, the intensity of which could be determined directly from O-J-I-P kinetics; it is the turn-over number of $Q_A$ (**N**). The value of **N** was found to be directly proportional to the concentration of $HCO_3^-$ in solution (Figure 3), being maximal ($2.02 \times 10^5$ $s^{-1}$) at pH 6.5 in the buffer that was not depleted of $CO_2/HCO_3^-$. To test the possible correlation between **N** and the processes described by the $R_b$ and the 1-$R_c$, the $R_b$ was chosen because it was found to be the rate-limiting step of all reactions, involving $HCO_3^-$. The changes of the $R_b$ were directly proportional to the concentration of $HCO_3^-$ in the solution (Figure 3). To test the proportionality of obtained dependences, the $R_b$ and **N** were determined for reactions in solution 4, to which 3 mM of $HCO_3^-$ was added exogenously. Amounts of the $R_b$

and **N** (marked by asterisks in Figure 3) were very close to the straight lines describing the dependences of the $R_b$ and of **N** on the concentration of $HCO_3^-$. This indicates that the obtained dependencies are correct, at least for the $HCO_3^-$ concentration range of 5–365 μM. Therefore, the $R_b$ and the $1-R_c$ may reflect real photochemical process(es) involving $HCO_3^-$ in PSII (at least in the redox reactions of $Q_A$), since both **N** and the $R_b$ are directly proportional to the concentration of $HCO_3^-$ in solution.

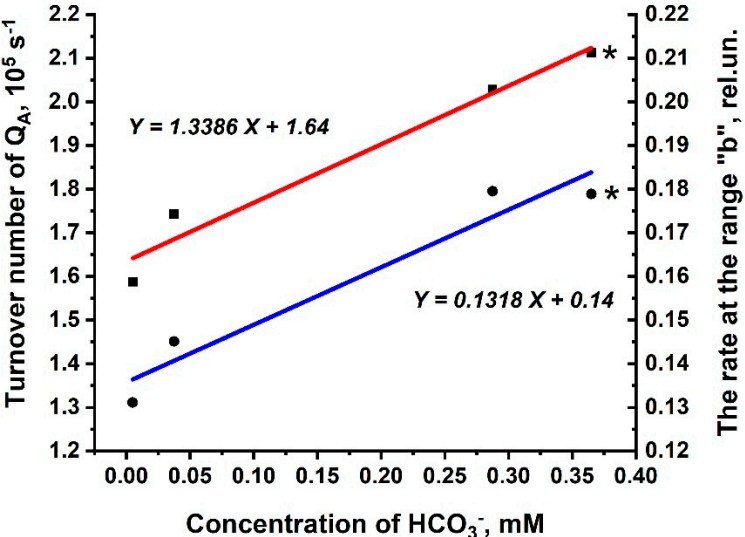

**Figure 3.** Turn-over number of $Q_A$ and the $R_b$ of O-J-I-P transients in BBY preparations as a function of $HCO_3^-$ concentration in the sample. The dependence of the turn-over number of $Q_A$ (**N**) on the $HCO_3^-$ concentration is shown by filled squares (■). Each point is an average of at least three separate experiments with calculated SD. Adj. R-Squares of line plotting is 0.95. The dependence of the rate at the range "b" (6–30 ms) of O-J-I-P transients ($R_b$) on the $HCO_3^-$ concentration is indicated by filled circles (●). Each point is an average of at least three separate experiments with calculated SD. Adj. R-Squares of line plotting is 0.91. Amounts of N and of the $R_b$ for BBY in solution 4, to which 3 mM of $HCO_3^-$ was added exogenously, are marked by asterisks. N and the $R_b$ were calculated using O-J-I-P kinetics, which was normalized to $F_0 = 0$ and to $F_P$ ($F_m$) = 1 (Figure 2B). The final concentration of BBYs was 10 μg Chl·mL$^{-1}$.

Using equations in Figure 3, a coefficient of proportionality (**k**) between N and the $R_b$ was calculated, being equal to $1.02 \times 10^6$ s$^{-1}$. This coefficient could be used in the following formula:

$$\mathbf{N} = \mathbf{k} \times R_b \tag{11}$$

to determine the rate of utilizing $HCO_3^-$ in the processes associated with the range "b", if only **N** is known or vice versa.

Thus, using kinetic parameters, derived from the rates of fluorescence rise in O-J-I-P transients, two distinct photochemical processes, dependent on $HCO_3^-$, have been revealed in PSII.

## 4. Discussion

The results obtained here clearly indicate that the removal of BC: (i) affects electron transfer in PSII; (ii) influences both the single turn-over and multiple turn-over events; (iii) simultaneously decreases electron transfer rate at two independent sites in PSII, which have different kinetic parameters for involving BC. However, further questions are: (1) Where does BC act? (2) How BC may be involved in photochemical events? Below possible sites of BC action will be discussed, and the possible mechanisms of its involvement will be presented.

*4.1. HCO$_3^-$ May Be Involved in a Single Turn-Over Event*

A single turn-over event was proposed to be associated with the O-J part of the O-J-I-P kinetic curve [33]. It is expected that single turn-over events are reflected not only by the R$_a$, but also by $\tau$F$_j$, F$_j$, and 1-F$_j$, since F$_j$ also belongs to the O-J part. What processes could be associated with the O-J rise? The description of these biophysical processes is a very complicated problem, since several processes have been proposed, which contradict each other on some issues (for a review, see [47]). Nevertheless, here an attempt will be made to explain the involvement of BC in single turn-over events.

4.1.1. The Shift of the $\tau$F$_j$

It is known that the shift of F$_j$ to the lower times led to an increase in the fraction of PSII with Q$_A$-Q$_B$ state [37]. Here, an opposite effect has been found, i.e., $\tau$F$_j$ shifted to the higher times, under the removal of BC (Table 2). This indicates the decrease in the fraction of PSII with the Q$_A$-Q$_B$ state, which may be due to the block of electron transfer between the donor side and Q$_A$. Thus, the decrease in HCO$_3^-$ concentration, reflected in the shift of $\tau$F$_j$ to the higher times, may induce the inhibition of electron transfer from the donor side to Q$_A$.

4.1.2. The Increase in F$_j$ or the Decrease in 1-F$_j$

The explanation for changes in the level of F$_j$ is ambiguous. On the one hand, the F$_j$ level depends on the availability of oxidized PQ molecules in the Q$_B$ site [36,48], but, on the other hand, the F$_j$ level, to some extent, depends on the presence of oxidized primary donor of PSII (P680), the amount of which, depends, in turn, on the limitation of electron transfer from the electron donor side of PSII to P680 [37,49,50]. Therefore, the removal of HCO$_3^-$, which was reflected (in our experiments) by the increase in F$_j$, may simultaneously inhibit two processes: (i) the influx of electrons from the donor side to Q$_A$, (ii) the outflux of electrons from the reduced Q$_A$ to Q$_B$ and further to the PQ pool.

The decrease in 1-F$_j$ indicates the inhibition of electron transfer from Q$_A$ since the parameter 1-F$_j$ reflects the probability of electrons moving into the electron transfer chain further than the Q$_A$ [33]. Since a decrease in 1-F$_j$ is accompanied by the decrease in HCO$_3^-$ concentration, HCO$_3^-$ deficiency may slow down the electron transfer from Q$_A$, which supports the involvement of HCO$_3^-$ into the outflux of electrons from Q$_A$ to the PQ pool suggested above. Thus, the parameters F$_j$ and 1-F$_j$ reflect the balance of electron flow at Q$_A$, which depends on the presence of HCO$_3^-$ ions.

An interesting information can be obtained from a comparison of magnitude of the bicarbonate effects on F$_j$ and 1-F$_j$. Taking into account that F$_j$ reflects the balance between the influx and the outflux of electrons at Q$_A$, while 1-F$_j$ reflects only the outflux of electrons from Q$_A$, the impact of the influx of electrons to Q$_A$ (i.e., donor side effect) may be calculated simply by subtracting the magnitude of BC effect on 1-F$_j$ from the magnitude of the effect on F$_j$. This effect seems to be 10% for thylakoids and 10–15% for the BBYs. This is close to the BC effect (~20%) on electron transfer on the donor side obtained on BBYs (see [30]), which confirms that the impact of the BC effect on F$_j$ (equal to 10–15%) may be related to the events on the donor side of PSII. Therefore, BC effects on F$_j$ are distributed as follows: 10% on the donor side and 40% on the acceptor side for thylakoids and 10–15% and 10%, respectively, for BBY. Thus, the balance of involvement of HCO$_3^-$ in single turn-over events shift to the acceptor side in thylakoids, but it shifts to the donor side in BBYs. Possible reasons for this phenomenon will be discussed below.

The differences in the involvement of HCO$_3^-$ in single turn-over events found for thylakoids and for BBY seem to be due to different concentrations of oxidized Q$_B$ in these preparations before the experiment starts, i.e., in darkness. Actually, it is known that thylakoids contain 6–12 PQ molecules per RCII [37], while BBY appears to have only ~2 PQ molecules per RCII in cyanobacteria [17]. Therefore, in single turn-over events, the low number of oxidized Q$_B$ in BBY actually may induce the shift of involvement of HCO$_3^-$ to the donor side, whereas, a larger number of PQ molecules in thylakoids

may shift the involvement of $HCO_3^-$ to the acceptor side. The shift of this balance could explain the opposite BC effect on the 1-$R_a$ in thylakoids and in BBY (cf. Figures 1A and 2A; Tables 2 and 3, column 6). The existence of the BC-mediated balance between oxidation and reduction of $Q_A$ indicates that BC effects on the acceptor and on the donor sides may be interconnected. However, the mechanism of this interaction is currently unclear.

The $R_a$, being a part of the O-J rise, may reflect the total BC effect on both sides of PSII, since the magnitude of the BC effect was equal to ~30% (Tables 2 and 3, column 6), which is ~2 times higher than the impact of the donor side in BC effect on $F_j$ (see above). Thus, the $R_a$ seems to be unsuitable for characterizing BC effects on different sides of PSII on chloroplasts and on leaves. In contrast, $\tau F_j$, $F_j$, and 1-$F_j$ have the potential to study the effects of BC deficiency on PSII in leaves, since they allow us to separate the BC effects on the donor side from those on the acceptor side.

Summarizing the above, the following proposal is presented. $HCO_3^-$ is involved in the formation of the redox state of $Q_A$, which is reflected in the O-J part of O-J-I-P kinetics. The redox state of $Q_A$ is the result of a balance of at least two or more biophysical processes, and single turn-over events, involving $HCO_3^-$. These processes appear to be: (i) the electron transfer from the donor side to P680 and further to $Q_A$, (ii) the electron transfer from $Q_A$ to the PQ pool. Thus, the decrease in $HCO_3^-$ concentration induces the inhibition of single turn-over events, taking part both near $Q_A$ and on the donor side of PSII.

### 4.2. $HCO_3^-$ May Be Involved in Multiple Turn-Over Events

The change of the J-I-P phase under the removal of BC (Figures 1A and 2A) seems to be associated with multiple turn-over events, which indicates that BC is involved in dynamical photochemical processes. Moreover, it seems that BC takes part in two different photochemical processes because the changes in O-J-I-P kinetics are observed in its different parts: J-I rise and I-P rise (see Figures 1A and 2A). The difference between these processes is supported by the difference in their kinetic parameters based on $HCO_3^-$ utilization (see Table 4). We ask: could we use the kinetic parameters of enzymatic reactions to describe the use of BC in PSII? Yes, we can, since Khanna et al. [51] and van Rensen and Vermaas [52] have shown that the BC effect on photosynthetic $O_2$ evolution obeys the principles of enzymatic kinetics. Nevertheless, $K_m$ found in these works had reflected whole electron flow in PSII from water to ferricyanide, while here $K_m$ was calculated for two different processes. Thus, in this work, we describe for the first time the kinetic parameters of the involvement of bicarbonate in at least two dynamical photoinduced processes in PSII. Further, we will discuss where these dynamical processes take place.

### 4.2.1. The Changes in J-I Rise May Be Associated with Events on the Acceptor Side of PSII Resulting from Removal of $HCO_3^-$

What dynamical biophysical process can be associated with the change in the J-I rise (the $R_b$ in Figures 1A and 2A)? On the one hand, the J-I rise has been proposed to describe the efficiency of the electron transfer from $Q_A$ to $Q_B$ and the removal of the reduced $Q_B$ to the PQ pool [37]. On the other hand, BC, being located near the non-heme Fe, has also been found to facilitate the electron transfer from $Q_A$ to $Q_B$ [5,7,53]. Therefore, the removal of BC, decreasing the $R_b$ in J-I rise (Figure 1A), may inhibit at least the electron transfer from $Q_A$ to $Q_B$ and, probably, the replenishment of the PQ pool by plastoquinol. The involvement of BC in these dynamical processes is supported by the finding that the turn-over of $Q_A$ is directly proportional to the rate of J-I rise and, moreover, it is directly proportional to the concentration of BC in solution (Figure 3). Remarkably, the BC effect on the $R_b$ (see Table 2, column 8), being equal to ~80%, is similar to the BC effect (equal to ~85%) observed earlier, which was found to be associated with the electron transfer from $Q_A$ into the electron transfer chain on the acceptor side of PSII [54]. Thus, the BC effect on the $R_b$ may be associated both with the electron transfer from $Q_A$ to $Q_B$ and with the transfer of plastoquinol to the PQ pool.

How could BC increase the rate of electron transfer from $Q_A$ to $Q_B$? It is known that BC stabilizes electron transfer from $Q_A$ to $Q_B$, being tightly bound near the non-heme Fe [5]. Nevertheless, the detailed mechanism of this stabilization needs further investigation.

How could BC facilitate the movement of plastoquinol into the thylakoid membrane? The movement of plastoquinol into the PQ pool seems to depend on: (i) the rate of the reduction in $Q_B$ by $Q_A^-$, and (ii) the rate of protonation of $Q_B$. BC, being the donor of protons, was found to be involved in the protonation of $Q_B$ [53] (for additional information see [5]). During the protonation of $Q_B$, $HCO_3^-$ is transformed to $CO_3^{2-}$. To get ready for new protonation, $CO_3^{2-}$ received $H^+$ s from the stroma. Therefore, the fast protonation of $Q_B$ by BC may determine the rate of the removal of plastoquinol into the thylakoid membrane.

The above hypothesis (cf. [5]) is based on the assumption that BC is tightly bound on the acceptor side of PSII. If BC is tightly bound, it is expected that the rate of protonation of $Q_B$ (and, consequently, the rate of electron transfer) should depend only on $H^+$ concentration in the buffer, but not on the concentration of BC in this buffer (if 1 $HCO_3^-$ is associated with 1 RCII of PSII, as indicated in X-ray structures [14,15,17]). In contrast, in this work, the electron transfer rate (reflected in the $R_b$) was found to be dependent on BC concentration (see Figure 3). Remarkably, $H^+$ concentration did not affect electron transfer because pH was constant in solutions 2, 3, and 4 (see above) due to the high concentration and high buffer capacity of the buffer used (for more details see "Materials and Methods"). To explain the above discrepancy, it was assumed that BC may be the constantly exchangeable ligand near the non-heme Fe. According to this new proposal, $H^+$ s may originate not from stroma, but from $H_2CO_3$, which is formed from $CO_2$ and $H_2O$ near the non-heme Fe. Further, $H^+$ transfers from $H_2CO_3$ to the $Q_B$ forming $HCO_3^-$ and then this $HCO_3^-$, we suggest, is replaced by a new molecule of $H_2CO_3$. This cycle, repeating over and over, could provide the constant protonation of $Q_B$. This hypothesis is in agreement with the work of Brinkert et al. [18], where the binding constant of $HCO_3^-$ at non-heme Fe was found to be changed during PSII function, which indicates that BC could dissociate from its binding site. Thus, $HCO_3^-$ may take part in the protonation of $Q_B$, as a constantly exchangeable substrate. However, this hypothesis needs additional examination.

Remarkably, $k_{cat}$ for $R_b$ (which may reflect the rate of protonation of $Q_B$) is about an order of magnitude lower than $k_{cat}$ for 1-$R_c$. This indicates that the involvement of $HCO_3^-$ in the protonation of $Q_B$ may be the rate-limiting step in electron transfer through PSII. This assumption is supported by previous work where authors have concluded that "the rate-limiting step in electron transfer in $HCO_3^-$ depleted thylakoids may be the protonation of $Q_B^-$ and possibly $Q_B^{2-}$" [8]. Thus, the involvement of $HCO_3^-$ in the protonation of $Q_B^-$ or $Q_B^{2-}$ may be the rate-limiting step of all photochemical reactions, taking part in PSII.

### 4.2.2. The Changes in I-P Rise May Be Associated with Events on the Donor Side of PSII Resulting from Removal of $HCO_3^-$

We now ask: where could be the place where the dynamical biophysical process, described by the change in I-P rise (the 1-$R_c$ in Figures 1A and 2A), takes place? On the one hand, it is known that the I-P rise may reflect the rate of electron transfer from PSII to PSI through the PQ pool and cytochrome $b_6f$ complex (for more details see reviews [34,37]). On the other hand, the value of $F_P$ (the peak of I-P rise), being equal to $F_m$, may reflect the functional state of the donor side of the PSII [10,29,30]. Thylakoid membrane preparations contain almost all components of the electron transfer chain of thylakoids. Therefore, the I-P rise in thylakoids, we suggest, describes both the function of the donor side of PSII and the electron transfer from PSII to PSI.

BBYs, containing almost only PSII, may help to further study the BC effect only in PSII and mainly on its donor side. Actually, in BBY, a clear BC effect was observed reflected in the decrease in $F_P$ and of the 1-$R_c$ under the removal of $HCO_3^-$ (Figure 2A). This effect is thought to be associated with the donor side based on the following facts: (1) the decrease in $F_m$ ($F_P$) has been repeatedly shown under the removal of BC that has proved to be associated with the inhibition of electron transfer on the donor side of PSII [10,12],

especially, this decrease has been found to become stronger in the presence of inhibitors of carbonic anhydrases [29,30]; (2) the magnitude of the BC effect on the 1-$R_c$ (equal to 6% (Table 3, column 9)) and, especially, on $F_P$ (equal to 34% (Table 3, column 5)) was similar to that, which has been observed on the donor side of PSII by the rate of photo-induced $\Delta F$ growth and by photosynthetic $O_2$ evolution (the magnitude was found to be within the range of 22–23% [30]), compared to the values (in the range of ~86% or 6–7 times) of BC effects associated with the acceptor side [54]. Therefore, the $F_P$ level and the 1-$R_c$ may be the diagnostic feature of the donor side inhibition when BC is removed from the BBY= preparations.

Could $F_P$ and 1-$R_c$ be the universal indicator of inhibition of the donor side for both BBY and thylakoids? Remarkably, both 1-$R_c$ and $F_P$ showed the effect of BC, which was more pronounced in BBY, compared to that in the thylakoids. Nevertheless, the removal of BC step by step resulted in the decrease step by step of only the 1-$R_c$ in both the preparations, while $F_P$ changed in the same manner only in a limited range of concentration of $HCO_3^-$ (above 38 µM (see Table 2, column 5). This indicates that only the parameter 1-$R_c$ seems to be the same for both samples. The changes in the 1-$R_c$ are statistically significant (despite the low difference in percentage). Taking into account that JIP test clearly resolves the changes in 1-$R_c$ (Figures 1A and 2A), we conclude that the 1-$R_c$ can be used for diagnostic purposes in all types of preparations, even in a leaf.

Summarizing the above data, the changes in I-P rise, especially the changes in the 1-$R_c$, when $CO_2/HCO_3^-$ is removed, may be associated with the inhibition of electron transfer on the donor side of PSII. Taking into account that the parameter 1-$R_c$ characterizes actually the rate of some photochemical process(es) and that these photochemical process(es) belong to multi-turn-over events, it is assumed that this process(es) should be dynamical. This indicates that $HCO_3^-$ is actually involved in the dynamical processes on the (electron) donor side of PSII.

What dynamical biophysical processes on the donor side of PSII may involve bicarbonate? There is the hypothesis that BC may support photosynthetic $O_2$ evolution by "consuming" $H^+$ released during water oxidation to prevent the acidification of the area near the $Mn_4CaO_5$-cluster [19]. If BC "consumes" all $H^+$ s released, then PSII would need 4 $HCO_3^-$ per one $O_2$, since the evolution of one $O_2$ is accompanied by the release of 4 $H^+$s. Therefore, the rate of involving BC in $H^+$ consumption should be at least four times higher than the rate of $O_2$ evolution. $k_{cat}$ for 1-$R_c$ ($1.6 \times 10^7$) was found to be almost two orders of magnitude higher than the rate of $O_2$ evolution multiplied by 4 ($3.6 \times 10^5$ µmol $H^+$ (mole of RC)$^{-1} \cdot s^{-1}$), to find the rate of $H^+$ consumption by BC. This indicates that the consumption of $H^+$ s by $HCO_3^-$ may be the dynamical biophysical process, which is described by the 1-$R_c$.

### 4.3. Perspectives of Using the Obtained Data for Research on Plants in a Field

The findings, in this research, have revealed a set of parameters ("the fingerprint of BC effect"), the change of which can serve as an indicator of a change in the concentration of BC near PSII. Moreover, this "fingerprint" may be obtained on a leaf directly in a field, since the applicability of the non-invasive JIP test has proved to be very useful in this work. Thus, the results of this work have the potential to be used for the diagnosis of PSII state in agricultural plants in a field.

Many parameters that change by removing BC (for example J-I part, reflecting the $R_b$ in our work; 1-$V_j$, reflecting 1-$F_j$ in our work) were also previously recognized as indicators of stress conditions for plants [34,37]. It is possible that under stress conditions plants suffer not only from the stress factor itself but also from the lack of BC. On the other hand, a lack of BC might be an indicator of other types of stress. Thus, BC near the PSII may be a good diagnostic feature for the functional state of this complex. Nevertheless, further investigations are needed to test these hypotheses.

The coefficient **k** (see Section 3.3. of "Results") can allow one to quantitatively determine the amount of BC near PSII by measuring the $R_b$ or the turn-over number of $Q_A$.

It is possible that the amount of BC near PSII may correlate with the concentration of $CO_2/HCO_3^-$ near the centers of carboxylation in the stroma of chloroplasts. The efficiency of carboxylation determines the productivity of plants. Therefore, the amount of BC near PSII might be an indirect indicator of plant productivity. However, this assumption needs to be carefully examined using different methods. Thus, the data obtained here may be the basis for the development of new approaches to determine: (i) the productivity of plants in the field; (ii) is the plant under stress? (iii) what stress(s) is the plant currently under?

**5. Conclusions**

In this work, for the first time, $HCO_3^-$ was found to be simultaneously involved both in single turn-over and in multiple turn-over photochemical processes ("dynamical processes"). Remarkably, $HCO_3^-$ takes part in dynamical processes both on the (electron) acceptor and on the (electron) donor sides of PSII. What is more, the rate-limiting step of electron transfer and of the involvement of $HCO_3^-$ is on the acceptor side of PSII. Data, presented in this paper, imply that, on the acceptor side of PSII, $HCO_3^-$ may be involved in the protonation of $Q_B$, which is one of the dynamical processes. The involvement of $HCO_3^-$ in the removal of $H^+$ s, on the donor side, is also discussed as another dynamical process. We propose that the involvement of $HCO_3^-$ in dynamical processes is associated with the constant exchange of $HCO_3^-$ at its binding sites on both electron donor and acceptor sides of PSII. The changes, observed in the O-J-I-P fluorescence transients, may be used as a diagnostic property in future use for in vivo experiments, revealing the inhibition of PSII by the deficiency of $HCO_3$. This will be especially useful if there is confidence that the photosynthetic organism is under the condition of deficiency of inorganic forms of carbon.

**Supplementary Materials:** The following supporting information can be downloaded at: https://www.mdpi.com/article/10.3390/photochem2030050/s1, Figure S1: The Lineweaver and Burk plot for the rate $R_b$ (6–30 ms) of O-J-I-P transient vs the substrate ($HCO_3^-$) concentration, obtained for thylakoids preparation; Figure S2: The Lineweaver and Burk plot for the rate $1$-$R_c$ (400–800 ms) of O-J-I-P transient vs the substrate ($HCO_3^-$) concentration, obtained for thylakoids preparation; Figure S3: The Lineweaver and Burk plot for the rate $R_b$ (6–30 ms) of O-J-I-P transient vs the substrate ($HCO_3^-$) concentration, obtained for BBY preparation; Figure S4: The Lineweaver and Burk plot for the rate $1$-$R_c$ (400–800 ms) of O-J-I-P transient vs the substrate ($HCO_3^-$) concentration, obtained for BBY preparation.

**Funding:** Financial support was provided by the Ministry of Education and Science of the Russian Federation (themes FMRM-2022-0012 and 121040600140-3).

**Institutional Review Board Statement:** Not applicable.

**Informed Consent Statement:** Not applicable.

**Data Availability Statement:** Not applicable.

**Acknowledgments:** I dedicate this work to dear Govindjee Govindjee, who is celebrating his 90th birthday on 24 October 2022. I thank Govindjee Govindjee for valuable discussions on the role of bicarbonate in PSII function, V.V. Terentyev for assistance in maintaining the equipment for the removal of $HCO_3^-$ from solutions, and for help in determining $HCO_3^-$ concentrations in solutions. The study was performed using the equipment of the Shared Core Facilities of the Pushchino Scientific Center for Biological Research (http://www.ckp-rf.ru/ckp/670266/, accessed on 1 August 2022).

**Conflicts of Interest:** The authors declare no conflict of interest.

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
