# Peer review of "An Insight into the Bicarbonate Effect in Photosystem II through the Prism of the JIP Test"

_2673-7256, doi:10.3390/photochem2030050_

Round 1

Reviewer 1 Report

An extensive revision is needed.

The text is written in FIRST person, using I----e.g. I report,--- I found etc.. This should be changed.

There are verbose sentences and redundant explanations. It can be concised.

There are a number of ABBREVIATIONS used. There can be separate section for abbreviations, which will make the manuscript readable and clear. Many are not clearly defined nor calculations for parameters are not shown in Materials & Methods. Example: Rb and Rc. 

Method for Chl estimation is bit obscure. What was the solvent and procedure be briefly mentioned.

Repeated use of a) or A in Fig.1 and Fig. 2 is confusing. Use different notations and delete (a) and (b) below the Figures.

Table 1 can be split into TWO TABLES separating Thylakoid and BBY particles be made.

Rewrite and resubmit.

Author Response

Thank you very much for reading this paper carefully and for the very helpful comments! I have presented my answers below point by point.

  • “An extensive revision is needed.” - The text has been significantly revised. Furthermore, I have sent the text to my colleague in USA. He is the native-speaker and has a rich experience in writing scientific articles and books.

  • “The text is written in FIRST person, using I----e.g. I report,--- I found etc.. This should be changed.” – This problem was fixed. My colleague suggests to change “I found etc.” to the “We found etc.” in some parts of the text. I agreed with him, since he is very experienced in style of academic English.

  • “There are verbose sentences and redundant explanations. It can be concised.” – The long sentences were concised when it was possible. The redundant explanations were shortened and sometimes deleted.

  • “There are a number of ABBREVIATIONS used. There can be separate section for abbreviations, which will make the manuscript readable and clear. Many are not clearly defined nor calculations for parameters are not shown in Materials & Methods. Example: Rb and Rc.” - The rules of the journal do not provide for a separate part reserved for abbreviations. But, agreeing with the relevance of the reviewer's remark, the abbreviations relating to the fluorescence parameters were combined into a table. Also, in this table are given quite detailed definitions of these parameters and methods for their calculation. This table was set in “Materials and Methods”.

  • “Method for Chl estimation is bit obscure. What was the solvent and procedure be briefly mentioned.” – Aqueous buffered acetone (80%) was used as the solvent. This information was added to the description of the method. The instrumental part of the spectrophotometric technique for measuring chlorophyll concentrations is the same for the vast majority of such methods, so I did not describe it. Moreover, it is very well described in the original article, to which the link is given.

  • “Repeated use of a) or A in Fig.1 and Fig. 2 is confusing. Use different notations and delete (a) and (b) below the Figures.” - (a) and (b) below the Figures were deleted from both Figures and text.

  • “Table 1 can be split into TWO TABLES separating Thylakoid and BBY particles be made.” - The Table 1 was divided into two.

Reviewer 2 Report

The author investigated the influence of the bicarbonate ion on the functions of photosystem 2, using chlorophyll a fluorescence induction (JIP test). Thylakoid membranes and BBY particles from peas are the object of the study.

Comments and remarks

In the introduction, the author describes in detail the functions of PS2 and the capabilities of the method used. It is necessary to accurately and clearly formulate your working hypothesis at the end of the introduction.

The first two subheadings in “Result” section (3.1 and 3.2) are more appropriate for the section “Materials and Methods”

The quality of the Table 1 and the Figures is necessary to improve.

The discussion should be shortened and written more tightly.

Author Response

Thank you very much for reading this paper carefully and for the very helpful comments! I have presented my answers below point by point

  • “In the introduction, the author describes in detail the functions of PS2 and the capabilities of the method used. It is necessary to accurately and clearly formulate your working hypothesis at the end of the introduction.” – The working hypothesis was formulated and added close to the end of “Introduction”

  • “The first two subheadings in “Result” section (3.1 and 3.2) are more appropriate for the section “Materials and Methods” – Sections 3.1 and 3.2 were changed and transferred to the section “Materials and Methods”

  • “The quality of the Table 1 and the Figures is necessary to improve.” - The Table 1 was divided into two. Figures were improved.

  • “The discussion should be shortened and written more tightly.” - The discussion was shortened and I tried to write it more compactly. I think, If I will reduce it further, some important information may be lost, or some of the conclusions will be incomprehensible without referring to the relevant literature sources.

Round 2

Reviewer 1 Report

The paper is adequately revised. This may be accepted in the present form.